# The ACL OCL Corpus: Advancing Open Science in Computational Linguistics

**Shaurya Rohatgi**[1*] **Yanxia Qin**[2*†] **Benjamin Aw**[2] **Niranjana Unnithan**[2‡] **Min-Yen Kan**[2]

[1] College of Information Sciences and Technology, Pennsylvania State University
[2]School of Computing, National University of Singapore
szr207@psu.edu, {qinyx, kanmy}@comp.nus.edu.sg

## Abstract

We present ACL OCL, a scholarly corpus derived from the **ACL** Anthology to assist **O**pen scientific research in the **C**omputational **L**inguistics domain. Integrating and enhancing the previous versions of the ACL Anthology, the ACL OCL contributes metadata, PDF files, citation graphs and additional structured full texts with sections, figures, and links to a large knowledge resource (Semantic Scholar). The ACL OCL spans seven decades, containing 73K papers, alongside 210K figures.

We spotlight how ACL OCL applies to observe trends in computational linguistics. By detecting paper topics with a supervised neural model, we note that interest in "*Syntax: Tagging, Chunking and Parsing*" is waning and "*Natural Language Generation*" is resurging. Our dataset is available from HuggingFace[1].

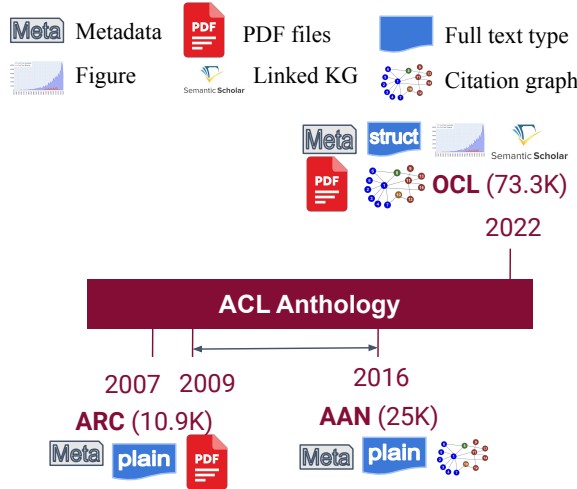

Figure 1: The ACL Anthology and the corpora built upon it. Compared to ARC and AAN, our OCL includes more data with additional structured full text, figures, and links to the Semantic Scholar academic graph.

## 1 Introduction

Building scholarly corpora for open research accelerates scientific progress and promotes reproducibility in research by providing researchers with accessible and standardized data resources. Driven by advancements in natural language processing and machine learning technologies, the computational linguistics (CL) discipline has experienced rapid growth in recent years. This rapid growth underpins the importance of having a scholarly corpus in the CL domain for ensuring sustainable progress.

The ACL Anthology[2] is an important resource that digitally archives all scientific papers in the CL domain, including metadata, PDF files, and supplementary materials. Previous scholarly corpora built on it, such as the Anthology Reference Corpus (ARC; Bird et al., 2008) and the Anthology

Author Network (AAN; Radev et al., 2009), extend its utility by providing full texts, citation and collaboration networks. However, both are becoming obsolete due to their outdated text extraction methods and insufficient updates.

We present the ACL OCL (or OCL for short), an enriched and contemporary scholarly corpus that builds upon the strengths of its predecessors while addressing their limitations. The OCL corpus includes 73,285 papers hosted by the ACL Anthology published from 1952 to September 2022. The OCL further provides higher-quality structured full texts for all papers, instead of previous string-formatted ones, enabling richer textual analyses. For instance, higher-quality full texts better foster the development of document-level information extraction tasks (Jain et al., 2020; Das et al., 2022). The structured information in full texts, such as sections and paragraphs, facilitates section-wise tasks such as related work generation (Hoang and Kan, 2010) and enables fine-grained linguistic analyses

---

*Equal contribution
†Correspondence author
‡Work done during internship in NUS.
[1] https://huggingface.co/datasets/WINGNUS/ACL-OCL
[2] https://aclanthology.org/

| Name | #Doc. | Text Type | Linked KG | Fig. | Peer | Source | Domain |
|------|-------|-----------|-----------|------|------|--------|--------|
| RefSeer (Huang et al., 2015) | 1.0M | string | CiteSeerX | × | partial | WWW | multi |
| S2ORC (Lo et al., 2020) | 8.1M | structured | S2AG | × | partial | multi | multi |
| CSL (Li et al., 2022) | 396K | — | self | × | all | CCJ | multi |
| unarXive (Saier et al., 2023) | 1.9M | structured | MAG | × | partial | arXiv | multi |
| ACL ARC (Bird et al., 2008) | 10.9K | string | self | × | all | ACL | CL |
| AAN (Radev et al., 2009) | 25K | string | self | × | all | ACL | CL |
| ACL OCL (Ours) | 73.3K | structured | S2AG | ✓ | all | ACL | CL |

Table 1: Comparison between ACL OCL and existing scholar corpora. Text type means the type of full text. "Peer" means whether the scientific document is peer-reviewed. N.B., S2ORC contains papers from multiple sources including arXiv, ACL (42K), and PMC. CCJ is short for Chinese Core Journal.

(Jiang et al., 2020a).

In addition, to advance multimodal research such as figure caption generation (Hsu et al., 2021), OCL extracts 210K figures from its source documents. Furthermore, we link OCL to large-scale scientific knowledge graphs to enrich OCL with external information. In particular, we consider information such as citation data from a larger scholarly database (e.g., Semantic Scholar) and linkage to other platforms (e.g., arXiv).

To showcase the scientific value of the OCL corpus, we illustrate its utility through a downstream application of temporal topic trends (Hall et al., 2008; Gollapalli and Li, 2015) within the CL domain. We first train a pre-trained language model (PLM) based topic classification method on a subset of 2,545 scientific papers with ground truth topic labels. We then extrapolate and integrate the model predictions as silver-labeled topic information in the OCL.

The contributions of this paper are as follows:

- We construct the ACL OCL, which augments the source ACL Anthology. The OCL provides structured full text for 73.3K CL papers, enriches metadata originated from Anthology by linking to an external knowledge graph, and extracts 210K figures. We also analyze the OCL, disclosing its statistics and the quality of full texts.

- We conduct a case study on OCL's temporal topic trends, showing the emergence of new topics like "Ethics" and witnessing the past glory of "Machine Translation". We validate the importance of supervised data in topic classification. Model-predicted silver topic labels are released together with OCL.

## 2 Related Work

Scholarly datasets typically fall into two categories: task-specific and open research-oriented. The former, designed to serve one task, includes selective information of scientific papers such as abstract and citation strings, paired with task-specific outputs such as entities (Hou et al., 2021), summaries (Cachola et al., 2020) and citation intent labels (Cohan et al., 2019). In contrast, open research-oriented scholarly datasets aim to provide comprehensive and fundamental information about scientific papers, such as metadata and full text. The open scholarly datasets not only facilitate researchers in refining their task-specific data but also aid in analyzing the characteristics of scientific papers or groups of them. Our work is in line with the open scholarly dataset construction to serve a wide range of applications.

Similar to other datasets, the OCL features publication metadata, a staple in open scholarly datasets. This can enhance metadata analysis and bibliographic research within the CL domain. A comparison of the OCL with existing datasets, excluding metadata aspects, is presented in Table 1. The OCL distinguishes itself from other corpora by the target domain, focusing on the computational linguistic domain same as ARC and AAN[3]. Their common source (i.e., ACL Anthology) provides higher-quality scientific papers, which are all peer-reviewed by domain experts. In contrast to them, the enriched and updated OCL corpus includes more papers and information.

Inspired by and following S2ORC, the OCL provides structured full texts with the scientific documents' discourse structure (i.e., sections), which enables more extensive textual analysis. In contrast to corpora that rely solely on internal papers for cita-

---

[3] https://clair.eecs.umich.edu/aan/index.php

tion networks and thus limit their completeness, the OCL is linked to a large knowledge graph to overcome the constraints. Furthermore, multi-modal features such as figures are extracted to foster research in document layout and multi-modality.

## 3  The ACL OCL Corpus

We start by crawling the PDF files and metadata from the source ACL Anthology. We then pass the PDF files through the full-text extraction process. We enhance this data using Semantic Scholar's API to fetch additional information about OCL papers.

### 3.1  Data Acquisition

From the ACL Anthology, we design crawlers to fetch all the PDF documents and metadata from its website. Meta information is included, as the website's version is more accurate than that obtained by PDF extraction, given its author-driven nature. We remove PDF files longer than 50 pages, which are mostly conference volume introductions[4] or handbooks. We then obtain 73,285 conference journal and workshop papers in the CL domain.

The dataset undergoes annual updates, during which we download and process papers recently added to the ACL Anthology. For these updates, we identify new additions by monitoring changes in URLs.

### 3.2  Full-text Extraction

Different from other platforms with easier-to-extract resources (e.g., LaTeX), the ACL Anthology only provides PDF files, which we can use to extract full texts for OCL. After an extensive comparison with open-source toolkits such as PDFBox[5] and pdfminer.six[6], we use GROBID[7] for the full-text extraction from PDF files. Our study validates findings from Meuschke et al. (2023) which found GROBID outperforms the other freely-available tools in metadata, reference, and general text extraction tasks from academic PDF documents. We take the S2ORC-JSON format used by Lo et al. (2020) and Wang et al. (2020) for our full-text schema, which includes complete information parsed from PDF files, such as metadata, authors, and body text with citations, references, sections and etc.

Due to the limitations of GROBID in formatting information extraction such as figures, we extract

| Field | Description |
|---|---|
| `acl_id` | ACL Anthology ID |
| `title` | Title of paper |
| `abstract` | Abstract from Anthology |
| `bib_key` | ACL bibliographic key |
| `pdf_hash` | SHA1 hash of PDF file |
| `plain_text` | Full text in string format |
| `full_text` | S2ORC-JSON object |
| `corpus_paper_id` | S2 corpus ID |
| `arXiv_paper_id` | arXiv paper ID |
| `citation_count` | Citation from S2 |
| `language` | Language written in |
| `predicted_topic` | Model-predicted CL topic |

Table 2: Simplified OCL schema, showing 12 of 27 fields. "S2" refers to Semantic Scholar. Associated figure schema not shown.

figures and tables from PDF files using PDFFigures (Clark and Divvala, 2016) following Karishma et al. (2023). Each extracted figure is associated with its caption texts, which show the figure ID and textual description in the paper. The figures are stored separately from the textual data in OCL.

### 3.3  Knowledge Graph Linking

We link the OCL corpus with knowledge graphs to enrich OCL with external information such as citations. We choose the recently released Semantic Scholar Academic Graph (S2AG, Kinney et al., 2023), which includes the most recent data. We use its Graph API[8] to connect an OCL document to its corresponding document in S2AG. While the S2AG Graph API offers general information such as metadata, authors, and citations, it does not provide full texts of the articles. This is where ACL OCL steps in to supply the complete structured full text, thereby expanding the range of information and providing opportunities for more analysis.

**Citation Network.**  Different from the previous AAN corpus, we construct the citation network using the citation information from both inside and outside of OCL. Following AAN, we use in-degree to denote the number of incoming citations internal to OCL. We use citations to mention the number of references to OCL articles beyond OCL. In particular, we use the *citationCount* field provided by the S2AG Graph Paper Lookup API via ACL Anthology ID for external citation information. In

---

[4] https://aclanthology.org/2022.acl-long.0.pdf
[5] https://pdfbox.apache.org/
[6] https://github.com/pdfminer/pdfminer.six
[7] https://github.com/kermitt2/grobid

[8] https://api.semanticscholar.org/api-docs/graph

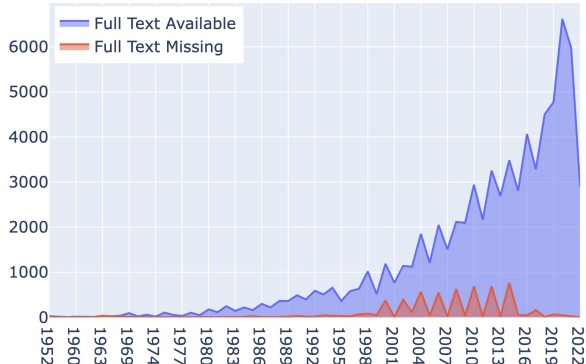

Figure 2: Growth in publication rates in the OCL. We observe that our full-text extraction fail mostly for papers between the years 2000 to 2015.

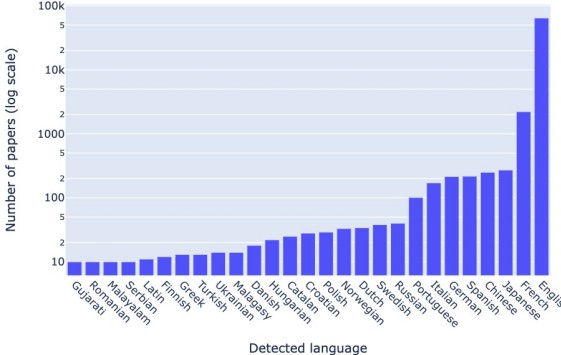

Figure 3: Distribution of the top 27 languages processed in the OCL corpus. Note that the scale of the y-axis is logarithmic.

total, we have 669,650 internal directed connections among the 73K papers in ACL OCL.

## 3.4 Data Schema

Our dataset adheres to the standard Semantic Scholar schema[9] (c.f., Table 2; complete listing in Appendix A) that resembles scientific documents, ensuring an organized and consistent structure. We note that certain fields are only valid on a portion of the papers. For example, only 16.8% of OCL papers have `arxiv_paper_id`, indicating a corresponding version in arXiv.org.

We add automatically detected fields such as `language` and `topic` to enable further analysis. To detect the language in which a document is written, we utilize a public language detection toolkit[10] (Shuyo, 2010) with its title and abstract as inputs. We discuss research topic detection next in Sections 5 and 6.1.

In addition to the standard CSV format, we provide our dataset in Apache Parquet, an open-source, column-oriented data format, to facilitate efficient downstream processing.

## 4 Dataset Analysis

We present statistical analyses of the OCL corpus, including the distribution of papers across years and linguistic distribution. We further highlight the quality of full texts and citation graph analysis.

### 4.1 Statistics

**Annual Publication Growth.** The annual quantity of published papers is computed and presented in Figure 2. The trend of annual publications shows an exponential growth pattern, especially after the year 2000. This escalating trend indicates the CL community is experiencing a rapid era of development, which underscores the need for resources like OCL to manage and leverage this rapidly expanding knowledge base. The noticeable spike in publications during even years, relative to odd years, can be primarily attributed to the scheduling of certain conferences that exclusively occurred in even years, such as COLING.

**Linguistic Distribution.** We also investigate the distribution of written languages of CL publications. Figure 3 illustrates the distribution of the top 27 out of 96 languages within the corpus. As expected, English is the most dominant language processed in the ACL OCL. However, it is noteworthy that languages such as Latin and Gujarati are also present. Prior research (Ranathunga and de Silva, 2022) on the representation of low-resource languages in the ACL Anthology found that numerous languages remain underrepresented. Even among languages in the same language group, there is a significant disparity in coverage.

### 4.2 Full-text Quality

Although we utilize an accurate full-text extraction toolkit GROBID, its inherent limitations do influence the quality of full texts in OCL. We manually

---

[9] https://api.semanticscholar.org/api-docs/.
[10] https://github.com/Mimino666/langdetect

| Title | $R_{deg}$ | $R_{cit}$ | Diff | Year | Topic |
|---|---|---|---|---|---|
| BERT: Pre-training of Deep Bidirectional Transformers for Language ... | 1 | 1 | 0 | **2019** | ML |
| Bleu: a Method for Automatic Evaluation of Machine Translation | 2 | 3 | 1 | 2002 | Summ |
| GloVe: Global Vectors for Word Representation | 3 | 2 | -1 | **2014** | LexSem |
| Moses: Open Source Toolkit for Statistical Machine Translation | 4 | 14 | 10 | 2007 | MT |
| Deep Contextualized Word Representations | 5 | 8 | 3 | **2018** | LexSem |
| Building a Large Annotated Corpus of English: The Penn Treebank | 6 | 9 | 3 | 1993 | Syntax |
| Neural Machine Translation of Rare Words with Subword Units | 7 | 17 | 10 | **2016** | MT |
| A Systematic Comparison of Various Statistical Alignment Models | 8 | 19 | 11 | 2003 | MT |
| Minimum Error Rate Training in Statistical Machine Translation | 9 | 39 | 30 | 2003 | MT |
| The Mathematics of Statistical Machine Translation: Parameter Est... | 10 | $N$ | $N$ | 1993 | MT |
| Statistical Phrase-Based Translation | 11 | 25 | 14 | 2003 | MT |
| Learning Phrase Representations using RNN Encoder-Decoder for ... | 12 | 4 | -8 | **2014** | MT |
| ROUGE: A Package for Automatic Evaluation of Summaries | 13 | 10 | -3 | 2004 | Summ |
| Enriching Word Vectors with Subword Information | 14 | 11 | -3 | **2017** | LexSem |
| Convolutional Neural Networks for Sentence Classification | 15 | 6 | **-9** | **2014** | ML |

Table 3: Top 15 OCL papers ranked by in-degree ($R_{deg}$), together with their citation ranking ($R_{cit}$), the ranking discrepancy (Diff = $R_{cit} - R_{deg}$), publication year, and model-predicted silver CL topics. $N$ means a number larger than 50.

checked 21 documents to assess the quality of full texts, especially three aspects including metadata, general texts, and contents in specialized formats. Incorrect metadata such as author names exceeding three tokens, is a common issue. In response, the OCL uses the metadata from ACL Anthology. For general text extraction, some common issues include missing section names, merged paragraphs, the mixing of footnotes with texts, and misidentifying texts as table captions. Most of the above errors stem from the challenge of extracting formatting information from PDF files. As with other toolkits, GROBID also struggles with the extraction of tables and figures, especially in-line equations. Quantitative assessments are provided in Appendix D.

### 4.3 Citation Graph

Table 3 displays the most frequently cited OCL papers ranked by their in-degrees ($R_{deg}$), which refer to the internal citations within OCL. Citations from papers beyond OCL are denoted as citations ($R_{cit}$). We show the discrepancy between in-degrees and citations with their difference. By analyzing in-degrees and citations, we can gain insights into the research interests of communities besides CL and compare them with the priorities of the CL community itself. From Table 3, it is observed that seminal works in the CL domain such as Moses (Koehn et al., 2007) and minimum error rate (Och, 2003) are not influential outside of CL. On the other hand, the CNN for sentence classification (Kim, 2014)

and RNN Encoder–Decoder (Cho et al., 2014) are interesting contrasts. Out of these 15 top-ranked papers, there are 7 papers (bolded) published after 2010, indicating the most recent research interests in CL, namely neural models. By analyzing the research topics of these top-cited papers, machine translation (7 of 15) is still the dominant research topic in CL. Note that the system-predicted topics are silver data, which we detail later.

## 5 Objective Topic Classification

Topic information serves as a crucial attribute in retrieving scientific papers. We focus on objective topics (Prabhakaran et al., 2016), which are used to denote specific research tasks such as machine translation or text generation. Notably, even though authors submit topics during manuscript submission, this information remains invisible on the ACL Anthology website. We aim to assign the most appropriate objective topic to each CL paper and further explore how this topic information can benefit the development of the CL community. The single-label topic classification setting is adopted in this paper for simplicity; multi-label classification is left to future work.

Given a scientific document $d$, with its textual information such as title, abstract, and full text, objective topic classification aims to assign a task topic label $l \in L$ to $d$. $L$ is the topic label set taken from the submission topics (e.g., "Generation", "Question Answering") of the ACL confer-

ences. The complete list of 21 topics is presented in Appendix B.

Based on the amount of supervised information used for training, we explore three classes of methods for topic classification: unsupervised, semi-supervised, and supervised methods.

## 5.1 NLI-based Un- and Semi-supervised Methods

Given the absence of large-scale topic-labeled data, our initial investigation focuses on zero-shot document classification methods. Yin et al. (2019) fine-tuned BART (Lewis et al., 2020) on natural language inference datasets, thus achieving zero-shot prediction of many tasks including document classification. We follow their work and use a variant model BART-large-MNLI[11] to model the topic classification task as an inference task. To identify the topic label of a document $d$, the BART-large-MNLI model predicts the probability $p(l|d)$ that the hypothesis label $l$ is entailed from the premise document $d$. We denote this unsupervised method as BART-NLI-0shot.

Inspired by the label-partially-seen experimental settings in Yin et al. (2019), we establish a semi-supervised setup leveraging the limited labeled data (§6) for improved performance. Specifically, we fine-tuned the BART-large-MNLI model with the labeled data, which is tailored to fit the NLI task. We refer to this semi-supervised method as BART-NLI-FT.

## 5.2 Keyword-based Supervised Method

After obtaining over 2000 documents with ground-truth topic labels, we train a supervised model for topic classification. As salient information of documents, keywords are shown to be helpful for many tasks such as classification (Zhang et al., 2021), clustering (Chiu et al., 2020), summarization (Litvak and Last, 2008; Liu et al., 2021), etc. Inspired by these findings, we design keyword-based supervised methods for topic classification. Initially, keywords are extracted from each document of the training set. Subsequently, we select the top 10 topic-representative keywords[12] for each topic by TF-IDF. During inference, given a test document $d$, the topic containing the most matching topic-representative keywords in $d$ is considered

| Source | #Doc. | # Unique Topic |
|--------|-------|----------------|
| ACL 2020 | 705 | 21 |
| EMNLP 2020 | 681 | 19 |
| ACL-IJCNLP 2021 | 470 | 21 |
| EACL 2021 | 295 | 20 |
| NAACL 2021 | 394 | 21 |
| **Total** | 2545 | 21 |

Table 4: Statistics of the topic corpus, STop.

the most suitable topic. We explore different keyword extraction methods including TF-IDF and Yake! (Campos et al., 2020), both of which are simple and efficient.

## 5.3 PLM-based Supervised Method

Given the proven success of pre-trained language models (PLMs) in multiple NLP tasks, particularly with training data in a small scale, we utilize a PLM-based classification framework for our task. The framework employs a pre-trained language model to encode the input document and a softmax classification layer atop it for topic label prediction. In addition, we consider pre-trained language models trained from scientific documents, namely SciBERT (Maheshwari et al., 2021) and SPECTER (Cohan et al., 2020), to take advantage of their strong encoding power of domain-specific documents.

## 6 Experiments

**Topic Data Curation.** We crawl published papers of several online held CL conferences (e.g., ACL 2020, EACL 2021) between 2020 and 2022, together with their topics from those websites. After aligning those papers with the data in the ACL OCL, we obtained 2545 documents classified in 21 topics in total, present in Table 4. These documents together with their topics are used as our training and testing data. We use 5-fold cross-validation across all experiments, randomly selecting 2,036 (80%) papers balanced in each topic as our training set, and the remaining 509 (20%) as test. We use Macro F1, Weighted F1, and Accuracy (aka. Micro F1) as evaluation metrics for the multi-class topic classification. We rely on Accuracy to compare systems' performances.

### 6.1 Topic Classification

Table 5 shows that performance improves over the full range — from zero-shot, semi-supervised, su-

---

[11]https://huggingface.co/facebook/bart-large-mnli

[12]The number of keywords is selected from {10, 20, 30, 40} via a hyper-parameter test.

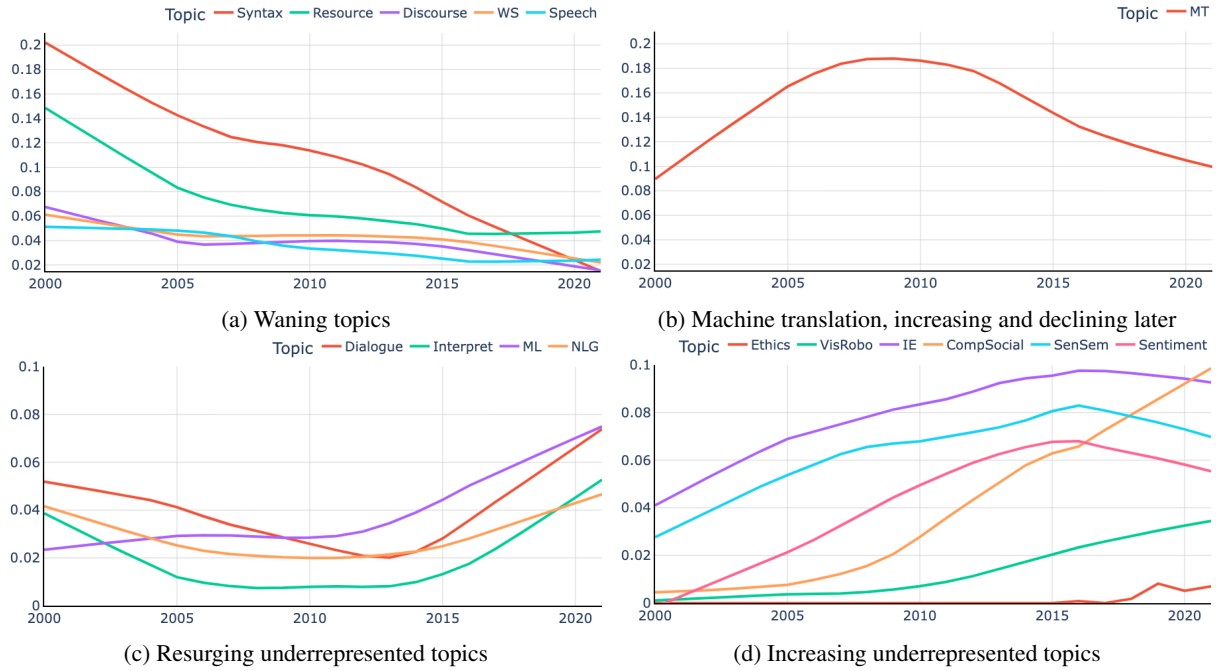

(a) Waning topics

(b) Machine translation, increasing and declining later

(c) Resurging underrepresented topics

(d) Increasing underrepresented topics

Figure 4: Plot of research trend of topics, grouped by patterns. The y-axis represents a topic's publication percentage in each year (x-axis). To identify the patterns, we remove the scattered data points and only show the topics' smoothed trending lines. Best viewed in color.

| Method | Mac. F1 | Wei. F1 | Acc. |
|---|---|---|---|
| BART-NLI-0shot | 38% | 43% | 45% |
| BART-NLI-FT | 44% | 50% | 51% |
| Keyword-TFIDF | 51% | 54% | 53% |
| Keyword-Yake! | 38% | 42% | 39% |
| PLM-SciBERT | 58% | 65% | 66% |
| PLM-SPECTER | 66% | 68% | 69% |

Table 5: Performances of topic classification models.

| Method | Abstract | I+C | Diff. |
|---|---|---|---|
| SciBERT | 66% | 67% | 1%↑ |
| SPECTER | 64% | 68% | 4%↑ |

Table 6: Performances (Accuracy) of different input texts, Abstract VS. I+C (Introduction+Conclusion).

pervised, to PLM-based methods. The trends highlight the importance of supervised data, even at small scales. The best performance (achieved by the PLM-based supervised method) is comparable to those reported in the state-of-the-art topic classification tasks on the scientific domain (Lo et al., 2020; Li et al., 2022). But the challenges of processing scientific data still remain, as performance is still much lower than those ($F_1 > 80\%$) in the news domain (Wang et al., 2022).

Inspired by (Meng et al., 2021), we explore

how different input text selection methods influence the task, namely Abstract and Introduction+Conclusion (Tabel 6). We adopt the I+C setting which has better performance.

**Case Study.** From the last column in Table 3 (system-predicted topics), we observe 13 correct labels out of 15 documents. Two works in *Resource & Evaluation*, namely BLEU (Papineni et al., 2002) and ROUGE (Lin, 2004), are incorrectly predicted due to insufficient training samples and high overlap with other topics. The 87% accuracy is higher than an expected 69% accuracy tested on the STop test set in Table 5, mainly because of the bias in the distribution of top-cited papers towards dominant topics (e.g., MT). Interestingly, three papers from lexical semantics including GloVe (Pennington et al., 2014) are correctly identified, perhaps due to the strong indication from words in their titles (i.e., "word representation" and "word vectors").

## 6.2 Topic Trend Analysis

We analyze the trend of model-predicted research topics in OCL starting from 2000 to 2021. Figure 4 presents the popularity (estimated by publication percentage) of all topics across years, subgrouped into recognizable trend patterns. We first introduce waning topics in Figure 4a, including both pre-

dominant ones like "Syntax" and "Resource" and underrepresented ones such as "Discourse" and "Speech". Another predominant topic in CL "MT", shown in Figure 4b, which peaks (around 19%) in the 2010s and declines in the latter years.

From all the remaining underrepresented topics, we further classify them into three types, only two types are shown in Figure 4. Figure 4c shows resurgent topics including "Dialogue", "Interpret", "ML" and "NLG", which have declining/low interests before 2015 but increased afterward. Figure 4d shows topics including "CompSocial", "IE", "Sentiment" and "SenSem", which are underrepresented historically and become noticeable later. Among them, "Ethics" is a very new and small topic starting in 2016. In contrast, "QA", "LexSem", "Summ" and "IR" (not shown) are relatively stable research topics with mild corrections in recent years. Among all topics, the popularity of "Syntax" drops the most (20%→2%) while "CompSocial" increases the most (1%→10%).

## 7 Downstream Applications

We previously highlighted one example application of ACL OCL, topic trend analysis. By nature, ACL OCL is a scholarly corpus aiming to enable and benefit a wide range of research. We now further depict a few existing research directions that could benefit from ACL OCL, predicting opportunities enabled by its characteristics.

- Additional forms of topic analyses and topic-aided tasks are now feasible with the domain-specific, fine-grained *topic information* provided by OCL, such as emerging topic detection (Asooja et al., 2016), evolution detection of topics (Uban et al., 2021), paper-reviewer matching (Thorn Jakobsen and Rogers, 2022) via topics and etc.

- OCL aids tasks that demand *complete textual content* from scientific papers, such as document-level terminology detection, coreference resolution, pre-training scientific large language models like Galactica (Taylor et al., 2022) or fine-tuning a general LLM like Llama (Touvron et al., 2023). Building scientific assistants for the CL domain, capable of responding to domain or task-specific questions (Lu et al., 2022) is also possible.

  *High-quality texts* are more suited for task-specific supervised data. For example, the abstracts and related work texts can be used as direct references for summarization and related work generation (Hoang and Kan, 2010; Hu and Wan, 2014) tasks, respectively.

- *Structures in full texts* such as sections and paragraphs provide opportunities for information/knowledge extraction tasks from specific sections, such as contribution extraction from Introductions and Conclusions, future work prediction from Conclusions, and analysis of generalization ability of NLP models[13] from Experiments.

- *Links to external platforms*, including arXiv, are beneficial as it allows us to access various versions of a paper in the ACL Anthology. As a result, this opens up opportunities for further analysis, such as examining the modifications made to the pre-print version of a paper before final publication, among other possibilities.

- As an up-to-date corpus *spanning decades*, OCL helps to analyze historical language change[14] in the CL domain, such as vocabulary change (Tahmasebi et al., 2021). To illustrate, consider the term 'prompt'. Traditionally in computer science, it referred to a signal on a computer screen (e.g., 'C:/>') that indicates the system is ready for user input. However, after 2020, its interpretation broadened to encompass a natural language instruction or query presented to an AI system (Jiang et al., 2020b). This newer definition is documented, for instance, in the Cambridge Dictionary[15].

- With *the combinations* of citation network, metadata, and full text, OCL can facilitate the construction of a knowledge graph, including papers, authors, topics, datasets, models, claims, and their relationships. Such a knowledge graph can enable a more intelligent academic search engine beyond keyword matching, facilitating information retrieval on multiple aspects.

- The *full texts and figures* provision multimodal research tasks such as summarization,

---

[13] https://genbench.org/workshop/
[14] https://www.changeiskey.org/event/2023-emnlp-lchange/
[15] https://dictionary.cambridge.org/dictionary/english/prompt

which integrates text summaries with figures and citation visualizations for a more holistic understanding.

# 8 Conclusion

We introduce ACL OCL, a scholarly corpus aiming to advance open research in the computational linguistics domain. The structured full texts, enriched metadata from Semantic Scholar Academic Graph, and figures provided by OCL can benefit existing research tasks as well as enable more opportunities. We highlight the utility of OCL in temporal topic trends analysis in the CL domain. The topics are generated by a trained neural model with a small yet effective scientific topic dataset. By analyzing the topics' popularity, we ask for more attention on the emerging new topics such as "Ethics of NLP" and the declining underrepresented topics like "Discourse and Pragmatics" and "Phonology, Morphology and Word Segmentation".

In the future, we will work on the data currency of OCL, which aims to keep OCL up-to-date with the ACL Anthology data. We plan to update OCL by year to keep it alive. The ultimate solution is to provide full-text extraction and information extraction APIs to ACL Anthology, thus hosting the OCL data on ACL Anthology itself.

# 9 Limitations

The OCL corpus is a small-scale collection of documents specifically focusing on peer-reviewed and open-access CL papers. As a result, it is not a comprehensive corpus like S2ORC (Lo et al., 2020), since it does not include any other sources beyond the ACL Anthology. In the future, the OCL could be expanded by incorporating CL papers on arXiv (e.g., cs.CL), which is related to unarXive (Saier et al., 2023). The challenge is how to filter out arXiv papers of low quality.

To ensure the extraction of high-quality full texts from the provided PDF files, the OCL corpus utilizes the most advanced open-sourced PDF2text toolkit, GROBID. Due to constraints on budget, only open-source toolkits are considered, although it is acknowledged that some paid PDF2text services might yield higher-quality full texts. In addition, previous work such as unarXive use LATEX files as source documents to avoid PDF2text.

The OCL corpus is an unlabeled resource, lacking annotations for specific tasks that require labels. Given the demonstrated capabilities of large language models (LLMs), we suggest that LLMs can play an instrumental role in generating high-quality, large-scale silver labels (Yu et al., 2023). Moreover, human-AI collaborative annotations (Liu et al., 2022) provide an effective strategy for complex tasks like natural language inference.

# Acknowledgements

This research is supported by the Singapore Ministry of Education Academic Research Fund Tier 1 (251RES2216, 251RES2037). We thank Sergey Feldman from Allen AI for providing Semantic Scholar data access.

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

## A Full Data Schema

The full data schema including 27 fields, is shown in Table 7.

| Field | Description |
| --- | --- |
| acl_id | Unique ACL Anthology ID |
| title | Title of paper |
| author | List of authors |
| abstract | Abstract from ACL Anthology |
| url | Publication link |
| year | Publication year |
| month | Publication month |
| booktitle | Publication venue |
| pages | Page range |
| address | Address of venue |
| doi | DOI number |
| journal | Journal name |
| volume | Volume number |
| number | Issue number |
| editor | Editor name |
| isbn | ISBN number |
| ENTITYTYPE | Publication type |
| bib_key | ACL bibliographic key |
| note | Notes from authors |
| pdf_hash | SHA1 hash of the PDF file |
| plain_text | Full text in string format |
| full_text | S2ORC-JSON object with sections |
| corpus_paper_id | S2 corpus ID |
| arXiv_paper_id | arXiv paper ID |
| citation_count | Number of citations from S2 |
| language | Language written in |
| predicted_topic | Model-predicted CL topic |

Table 7: Full data schema. S2 refers to Semantic Scholar.

## B  Topics in CL domain

We construct a taxonomy of objective topics in the CL domain, shown in Table 8, by taking and re-organizing the submission topics in major CL conferences (i.e., ACL, EMNLP, COLING). We surveyed their call for papers from 2000 to 2022, and include all topics in the taxonomy. We remove four broad coverage topics, including "Theme", "Student Research Workshop", "System Demonstrations", and "NLP Applications".

## C  Scientific Topic Dataset

We create the Scientific Topic Dataset, STop, by crawling scientific papers along with their topic labels from the following conferences:

- ACL 2020: https://virtual.acl2020.org/papers.html

- EMNLP 2020: https://virtual.2020.emnlp.org/papers.html

- ACL-IJCNLP 2021: https://2021.aclweb.org/program/overview/

- EACL 2021: https://www.virtual2021.eacl.org/index.html

- NAACL 2021: https://2021.naacl.org/conference-program/main/program.html.

## D  Text quality assessment

- Missing sections. 19 sections are not detected from 21 scientific papers. For example, "4.1 Analysis of Stemming", "Limitations, Conclusions, and Future Work".

- Footnote. All footnote numbers are concatenated to the body text, such as "Google Could Natural Language API1". There are 25 footnotes missing out of 92.

- Figures. There are 28.78% (19/66) figures not detected.

- Tables. There are 76 tables in total, 31.58% (24) of them are not detected and 46.06% (35) are detected but with either wrong contents or titles.

## E  Reading the corpus

The corpus file can be read and analyzed with the following Python commands:

```
df = pandas.read_parquet('ocl.parquet')
df.shape # for size
df.columns # for data schema
```

.

| ID | Abbr | Topic name |
|---|---|---|
| 1 | CompSocial | Computational Social Science and Social Media |
| 2 | Dialogue | Dialogue and Interactive Systems |
| 3 | Discourse | Discourse and Pragmatics |
| 4 | Ethics | Ethics and NLP |
| 5 | NLG | Generation |
| 6 | IE | Information Extraction |
| 7 | IR | Information Retrieval and Text Mining |
| 8 | Interpret | Interpretability and Analysis of Models for NLP |
| 9 | VisRobo | Language Grounding to Vision, Robotics and Beyond |
| 10 | LingTheory | Linguistic Theories, Cognitive Modeling and Psycholinguistics |
| 11 | ML | Machine Learning for NLP |
| 12 | MT | Machine Translation and Multilinguality |
| 13 | WS | Phonology, Morphology and Word Segmentation |
| 14 | QA | Question Answering |
| 15 | Resource | Resources and Evaluation |
| 16 | LexSem | Semantics: Lexical Semantics |
| 17 | SenSem | Semantics: Sentence-level Semantics, Textual Inference |
| 18 | Sentiment | Sentiment Analysis, Stylistic Analysis, and Argument Mining |
| 19 | Speech | Speech and Multimodality |
| 20 | Summ | Summarization |
| 21 | Syntax | Syntax: Tagging, Chunking and Parsing |

Table 8: Objective topics of papers in CL domain, together with defined abbreviations.