# OpenReview forum: "The ACL OCL Corpus: Advancing Open Science in Computational Linguistics"
_EMNLP/2023/Conference — EMNLP 2023 Main_

### Official Review · Reviewer_XBuc · 2023-08-01

**Soundness:** 3

**Excitement:**

3: Ambivalent: It has merits (e.g., it reports state-of-the-art results, the idea is nice), but there are key weaknesses (e.g., it describes incremental work), and it can significantly benefit from another round of revision. However, I won't object to accepting it if my co-reviewers champion it.

**Missing References:**

N/A.

**Paper Topic And Main Contributions:**

This very long (hard-to-read at times) paper provides three main contributions to the field of Computational Linguistics (CL). Firstly, it presents a new open source corpus (namely the ACL OCL corpus), based on the materials included in the ACL Anthology, enhancing this collection with "metadata [including the annotation of papers with their corresponding (main) topic, not present in the original resource], PDF files, citation graphs and additional structured full texts with sections, figures, and links to [...] Semantic Scholar". Secondly, some basic descriptive statistics methods are applied on the ACL OCL corpus in order to showcase its usefulness, i.e., by identifying trends in the topics being discussed in the papers in the ACL Anthology. And thirdly, it discusses potential scenarios where this new corpus could be applied for further research.

**Questions For The Authors:**

N/A.

**Reasons To Accept:**

The ACL OCL corpus, built by the authors is an interesting and potentially valuable resource for the CL community. Its topic labels can be quite useful in paper searches by topic within the ACL Anthology in case both resources are eventually and effectively interconnected.

**Reasons To Reject:**

The main contribution of the paper is the ACL OCL corpus and the anotation of papers with their topic(s). This could have been discussed in a short paper. The rest of the contributions of the paper are minor or look more like ongoing work. In particular, section 7 (Downstream Applications) are considered by this reviewer wishful thinking; the actual implementation of some of the proposals in this section would have meant a greater contribution to the field, up to the standards of an EMNLP long paper. Another interesting contribution would have been setting up the procedures to keep the corpus up-to-date, by means of a sustainable initiative. However, this is simply commented as future work in the paper.

**Reproducibility:**

3: Could reproduce the results with some difficulty. The settings of parameters are underspecified or subjectively determined; the training/evaluation data are not widely available.

**Reviewer Confidence:**

4: Quite sure. I tried to check the important points carefully. It's unlikely, though conceivable, that I missed something that should affect my ratings.

**Typos Grammar Style And Presentation Improvements:**

- Perhaps the statement "The ACL Anthology is the key resource that digitally archives *all* scientific papers in the CL 032 domain", in lines 030-032, should be conveniently hedged. There is life in CL beyond the ACL Anthology... Or so they say.

- There seems to be a typo in Table 1: the "k" in "73.3k" should be perhaps capitalized?

- There is space missing in line 147, between "handbooks." and "We"; and also in line 499, between "up" and "opportunities".

- In lines 198-199, the authors mention the total number of "internal directed connections among the 73K papers in ACL OCL". However, the number of external connections is missing. Mentioning this other number would be highly appreciated.

-  In line 202, the authors state that their "dataset adheres to a standard schema", but they fail to mention what standard it is, by whom it was developed, and/or where this standard can be retrieved.

- In line 338, there is an ill-formed inline citation, that is, "in (Yin et al., 2019)", which should be cited as "in Yin et al. (2019)" instead. Please, check the rest of the inline references in the paper, as there might be some other occurrences that need fixing this way.

- In line 471, " and" should be replaced by a comma (",").

- The discussion in lines 503-511 should be conveniently rephrased. To start with, the description of 'prompt' is quite arguable, and so is its description in relation with current AI agents querying.

- Thre seems to be a tpo in line 554 - "any" may have been meant instead of "and"?

---

> ### Author Rebuttal · Authors · 2023-08-29
>
> We extend our gratitude to reviewer XBuc for succinctly highlighting our key contributions. We are heartened by the acknowledgment of our resource's potential value and the utility of our topic labels for paper searches. A special thanks is for the meticulous identification of typos, ill-formed citations, overclaims, and ambiguities. We are committed to refining our draft based on these invaluable suggestions.
>
> We appreciate the reviewer's constructive feedback regarding the length and readability 'very long (hard-to-read at times)' of our draft. We are keen to make it more concise and reader-friendly. We recognize the challenges presented in lines 503-511 and are committed to revising it for clarity. We would be grateful for specific recommendations on which tedious section to condense.
>
> > Reject Reason A. *The main contribution of the paper is the ACL OCL corpus and the anotation of papers with their topic(s). This could have been discussed in a short paper. The rest of the contributions of the paper are minor or look more like ongoing work. In particular, section 7 (Downstream Applications) are considered by this reviewer wishful thinking; the actual implementation of some of the proposals in this section would have meant a greater contribution to the field, up to the standards of an EMNLP long paper.*
>
> We respectfully disagree with the reviewer's assertion that our work could be condensed into a short paper. The primary contribution, the ACL OCL corpus, is notably richer than previous offerings. Our structured full texts, figures, and links to Semantic Scholar were meticulously curated, offering significant potential for various downstream applications outlined in Section 7. Moreover, our topic labels for OCL papers provide an exemplar of how to derive valuable annotations from the corpus.
>
> While we acknowledge the primary focus on the corpus, we assert that the accompanying analyses, such as corpus analysis and trending topic analysis, hold inherent value. For instance, the highlighted exponential growth of annual publications not only signifies the community's progression but also underscores the need for sustainable development. This hints at broadening our research horizons to encompass documents outside our current purview, disseminate significant CL research, and encourage exploration into new or underrepresented topics.
>
> Section 7, "Downstream Applications", is intended to illuminate potential applications of the corpus. While it does not present fully-fledged results, it is pivotal in guiding future research trajectories. Predicting such potential directions is a hallmark of corpus studies.
>
> In conclusion, our work, spanning the comprehensive ACL OCL corpus, methodical topic detection, pertinent analyses, and proposed applications, collectively forms a substantial contribution to the CL domain. We believe this aligns well with the rigors of an EMNLP long paper.
>
>
> > Reject Reason B. *Another interesting contribution would have been setting up the procedures to keep the corpus up-to-date, by means of a sustainable initiative. However, this is simply commented as future work in the paper.*
>
> We recognize the importance of keeping the corpus current and appreciate the reviewer's emphasis on this aspect. To address this concern, we will incorporate the following strategies in the final version: (1) Open Sourcing the Code. By making our codebase publicly available, we facilitate transparency and encourage collaborative contributions from the research community. (2) Regular updates, including incrementally downloading new or changed data, processing it, and updating the repository. (3) Engaging the Community. We will actively seek community contributions to adding new papers, enriching metadata, or enhancing existing entries.
>
>
> > Typos Grammar Style and Presentation Improvements
>
> For the data schema, we adhere to the Semantic Scholar's standard schema, designed and maintained by their team. The schema can be accessed at https://api.semanticscholar.org/api-docs/.
>
> We explain the discussion in Lines 503-511 as follows: As a corpus spanning decades, OCL helps to study diachronic language change in the CL domain (Tahmasebi et al., 2021). To illustrate, consider the term 'prompt'. Traditionally in computer science, it referred to a signal on a computer screen (e.g., 'C:/>') that indicates the system is ready for user input. However, after 2020, its interpretation broadened to encompass a natural language instruction or query presented to an AI system [1]. This newer definition is documented, for instance, in the Cambridge Dictionary (https://dictionary.cambridge.org/dictionary/english/prompt).
>
>
> Reference:
>
> [1] Zhengbao Jiang, Frank F. Xu, Jun Araki, and Graham Neubig. 2020. How Can We Know What Language Models Know?. Transactions of the Association for Computational Linguistics, 8:423–438.

---

### Official Review · Reviewer_v8uW · 2023-08-01

**Soundness:** 4

**Excitement:**

4: Strong: This paper deepens the understanding of some phenomenon or lowers the barriers to an existing research direction.

**Missing References:**

David Jurgens, Srijan Kumar, Raine Hoover, Dan Mc-Farland, and Dan Jurafsky. 2018. Measuring the evolution of a scientific field through citation frames.Transactions of the Association for Computational Linguistics, 6:391–406.

**Paper Topic And Main Contributions:**

The paper presents an open corpus derived from the ACL Anthology to assist Open scientific research in the Computational Linguistics/NLP  domain as a replacement of the older ACL ARC and AAN corpora. It comprises metadata, PDF files, citation graphs and additional structured full texts with sections, figures, and links to a large knowledge resource (Semantic Scholar).
To demonstrate the usefulness of the corpus, the paper also reports an interesting case study on temporal research trends in the field.

**Questions For The Authors:**

A) Is the list of metadata you provide fixed, or will it be possible to include in the future additional metadata also referencing data or software provided?
B) You do mention the problem of extracting tables from pdf, but it remains unclear how you deal with tables in OCL. Can you please comment.
C) You also mention TF/IDF as a helpful method to extract possible topics. However, how do you deal with multi-word topics? Have you considered using terminology extraction tools to generate a first pool of possible topics?





**Reasons To Accept:**

The corpus comprises a state of the art textual resource based on the ACL Anthology (spanning almost 70 years of research in the field) with structured full texts, helpful metadata, links to the Semantic Scholar Academic Graph, and figures contained in the papers. It thus marks an important step forward in making available to the community the publication history of ACL papers.
The case studies are straightforward and well describe the usefulness, a.o., of citation graphs as well as the detection of research topics (based on unsupervised, semi-supervised, and supervised methods for topic classification).

**Reasons To Reject:**

I do not see strong reasons to reject the paper, neverthess I would recommend to include some references in the state-of-the-art section to
- scientometrics
- research data infrastructures, in particular the European CLARIN and the German text+ research data infrastructures on text and linguistic data
- metadata harvesting

**Reproducibility:**

5: Could easily reproduce the results.

**Reviewer Confidence:**

4: Quite sure. I tried to check the important points carefully. It's unlikely, though conceivable, that I missed something that should affect my ratings.

---

> ### Author Rebuttal · Authors · 2023-08-29
>
> We are heartened that reviewer v8uW recognizes our work as a significant advancement in providing valuable resources and analyses to the community. We are grateful for the insightful comments and the suggested references on research data infrastructures and measurement of the evolution of a field. We find resources in Common Language Resources and Technology Infrastructure (CLARIN) and German text+ are useful guidelines for designing metadata. We will enhance our work by incorporating the recommended references in the revised manuscript.
>
> > QA. *Is the list of metadata you provide fixed, or will it be possible to include in the future additional metadata also referencing data or software provided?*
>
> The current list of metadata provided is not rigid and can be easily extended. We understand the importance and benefits of including additional metadata, especially those referencing data or software resources related to scholarly work. To facilitate such updates, we can conveniently execute a pandas command. For example, to add a new field named 'lang_about', simply execute the command `metadata['lang-about'] = languages_list`.
>
>
> > QB. *You do mention the problem of extracting tables from pdf, but it remains unclear how you deal with tables in OCL. Can you please comment.*
>
> We acknowledge the challenges that arise from extracting tables directly from PDFs. As a result, we are currently working with a method wherein tables are included as images and metadata. This information is extracted using 'pdffigures', a tool that detects figures, tables, and other graphical material in scholarly documents. While this might not provide a text-based representation of table contents, it ensures the preservation and availability of table information for in-depth review. We continue to explore more efficient techniques for table extraction and hope to improve this feature in future iterations of the dataset.
>
>
> > QC. *You also mention TF/IDF as a helpful method to extract possible topics. However, how do you deal with multi-word topics? Have you considered using terminology extraction tools to generate a first pool of possible topics?*
>
> No, we did not use terminology extraction tools to generate a first pool of keywords, instead, we used a multi-word keyword extraction toolkit Yake! for this purpose.  We also consider both unigrams and bi-grams in TF-IDF based keyword extraction method to include multi-word keywords. We will revise Section 5.2 (Keyword-based Supervised Method) with more details.

---

### Official Review · Reviewer_s1m2 · 2023-08-03

**Soundness:** 3

**Excitement:**

3: Ambivalent: It has merits (e.g., it reports state-of-the-art results, the idea is nice), but there are key weaknesses (e.g., it describes incremental work), and it can significantly benefit from another round of revision. However, I won't object to accepting it if my co-reviewers champion it.

**Missing References:**

I think references are ok.

**Paper Topic And Main Contributions:**

The paper is about a corpus of texts from the ACL Anthology web site called Advancing Open Science
in Computational Linguistics (ACL OCL). Full texts are extracted from PDF files, together with internal structure (sections), metadata is added with automatic topic detection, citation graphs are also available, with links to Semantic Scholar. The corpus is available under open source licence (CC BY-NC 4.0) in Apache Parquet format.



**Questions For The Authors:**

Lines 144-145: Why is the version of metadata from the web site more accurate than that obtained by PDF extraction?
Lines 239-241 (Linguistic Distributeion): There is a difference between the paper being written IN a language and ABOUT a language, and both are interesting as metadata. Have you considered adding both types of metadata? In this respect the claim that "numerous languages remain underrepresented" is rather unclear. One can write about Icelandic in English or in Icelandic about English.
Line 250: When checking the 50 documents for text quality, the resulting claims are rather general - could you provide more accurate assessment (in numbers)?
Line 357: Is top 10 topic-representative keywords a number chosen arbitrarily or based on some tests?


**Reasons To Accept:**

I think ACL OCL is an important resource and should be presented at EMNLP 2023. As authors note the last similar effort was done some time ago and those corpora are now more or less obsolete, and the information extraction technology available then is outdated. It is positive that someone undertook a rather laborious corpus-building task which is perphaps of more infrastructural nature and less exciting research-wise but the result is quite valuable for reasons that are listed in the "Downstream Applications" section in the paper.

**Reasons To Reject:**

Corpus building in itself is not a very exciting topic any more and there is a feeling that work was done hastily to some extent - the authors admit themselves that they worked under budget constraints and used whatever was available freely (which is not a problem in itself). The other issue is the choice of Parquet format - I believe this rather limits the use of the corpus to the CL community. However, there is a large corpus linguistics community which usually uses CSV or other formats for corpora.

**Reproducibility:**

2: Would be hard pressed to reproduce the results. The contribution depends on data that are simply not available outside the author's institution or consortium; not enough details are provided.

**Reviewer Confidence:**

3: Pretty sure, but there's a chance I missed something. Although I have a good feel for this area in general, I did not carefully check the paper's details, e.g., the math, experimental design, or novelty.

**Typos Grammar Style And Presentation Improvements:**

Lines 111-114: I don't understand the grammar of the sentence - it would be better to rephrase.
Line 118: "same like" -> "same as"
Line 127: "more textual analysis" -> more extensive? (something is missing)
Line 146: mostlyconference -> mostly conference

---

> ### Author Rebuttal · Authors · 2023-08-29
>
> We thank the reviewer s1m2 for the thoughtful feedback and appreciation of our effort for this important resource. We are excited that the reviewer identified our motivation to build this resource and its value to advance 'Downstream Applications' as detailed in Section 7.
>
> > Reject Reason A. *Corpus building in itself is not a very exciting topic any more and there is a feeling that work was done hastily to some extent - the authors admit themselves that they worked under budget constraints and used whatever was available freely (which is not a problem in itself).*
>
> We agree with the statement that corpus building, while perhaps not seen as 'very exciting', remains 'valuable' and 'important', thus indispensable to the research community. We wish to clarify that the constraints on budget did not compromise the quality of our work. Our budgetary limitations indeed motivated us on some of our design choices, but these decisions were taken after careful consideration and not out of haste. For example, although we restricted ourselves to open-source toolkits for PDF text extraction, we undertook an evaluation of available options, eventually opting for the most advanced one. We would appreciate any specific examples of hasty decisions that you believe were made, so that we can address them appropriately.
>
>
> > Reject Reason B. *The other issue is the choice of Parquet format - I believe this rather limits the use of the corpus to the CL community. However, there is a large corpus linguistics community which usually uses CSV or other formats for corpora.*
>
> We acknowledge that Parquet is not as widely recognized as CSV or JSON. However, we opted for Parquet primarily because of its efficient compression and encoding, which reduces required storage space making it more accessible to researchers. Its columnar storage allows for faster query results and better compatibility with data processing tools like pandas.  In addition, Parquet files can be easily transformed to CSV files with a pandas command `parquet_data.to_csv('filename.csv')`. Although, we recognize CSV as a more popular option, thus we will offer a supplementary CSV file containing metadata for all 73K papers, while storing full texts in distinct files.
>
> > QA. *Lines 144-145: Why is the version of metadata from the web site more accurate than that obtained by PDF extraction?*
>
> The website-sourced metadata is more accurate because it is directly provided by the authors, ensuring its authenticity and precision. In contrast, extracting metadata from PDFs can sometimes introduce errors or miss out on certain data points. Thus, we take the website-sourced metadata in our corpus. We will clarify the rationale in Section 3.1 (Data Acquisition).
>
> > QB. *Lines 239-241 (Linguistic Distributeion): There is a difference between the paper being written IN a language and ABOUT a language, and both are interesting as metadata. Have you considered adding both types of metadata? In this respect the claim that "numerous languages remain underrepresented" is rather unclear. One can write about Icelandic in English or in Icelandic about English.*
>
> We appreciate this insightful suggestion of incorporating both written-IN and written-ABOUT languages. We will consider this as future work and note it in the conclusion section. The assertion that "numerous languages remain underrepresented" pertains specifically to the written-IN language.
>
> > QC. *Line 250: When checking the 50 documents for text quality, the resulting claims are rather general - could you provide more accurate assessment (in numbers)?*
>
> Thanks for the suggestion. We will add a detailed assessment in the appendix.
>
> > QD. *Line 357: Is top 10 topic-representative keywords a number chosen arbitrarily or based on some tests?*
>
> The selection was determined through a hyper-parameter search test from {10, 20, 30, 40}. We will add the details to the final version.
>
>
> > Typos Grammar Style And Presentation Improvements:
> *Lines 111-114: I don't understand the grammar of the sentence - it would be better to rephrase. Line 118: "same like" -> "same as" Line 127: "more textual analysis" -> more extensive? (something is missing) Line 146: mostlyconference -> mostly conference*
>
> We appreciate the reviewer's attention to detail in identifying the typos and offering writing suggestions.
> Lines 111-114 have been revised to: "Similar to other datasets, the OCL features publication metadata, a staple in open scholarly datasets. This can enhance metadata analysis and bibliographic research within the CL domain. A comparison of the OCL with existing datasets, excluding metadata aspects, is presented in Table 1." We will carefully revise the draft.

---

### Official Review · Reviewer_q3Sr · 2023-08-21

**Typos Grammar Style And Presentation Improvements:** line499
**Soundness:** 4

**Excitement:**

4: Strong: This paper deepens the understanding of some phenomenon or lowers the barriers to an existing research direction.

**Paper Topic And Main Contributions:**

The authors of this paper primarily address the limitations of available corpora containing scholarly articles in the computational linguistic domains such as ARC (Bird et al., 2008) and AAN (Radev et al., 2009). In doing so, they introduce ACL OCL, a corpus containing peer reviewed scholarly articles published in various computational linguistic conferences and workshops. The corpus contains 73K papers which is a significant step up from available resources (ARC and AAN).  In addition, the topical labels for a subset of the papers are also made available, which can aid in topic classification research. As an example use case of the corpus, the authors provide an analysis of the trends of research topics over the years (e.g., waning and resurgence). Furthermore, a number of future use cases of the corpus is also outlined.

**Questions For The Authors:**

1. Did you try making use of Large Language Models to generate the keywords instead of using TF-IDF and Yake?

**Reasons To Accept:**

1. The proposed corpus will be a valuable resource for the progress of research topics dealing with scientific text.
2. The structured data (e.g., title, abstract, section-wise full-text, figures, etc.) will be valuable for a multitude of research areas.
3. The corpus contains a large number of papers.
4. The data extraction/cleaning/curation methods seem very sound and completed with care.
5. The authors plan to maintain and regularly update the corpus.

**Reasons To Reject:**

My only concern is that this corpus will be a source of unlabeled data. While the authors do provide topic classification labels for a small subset, it is not extendable without a significant amount of human effort. As a result, it is unclear how easy/feasible/suitable it will be for downstream applications. However, providing labels for downstream tasks is probably out of scope for this paper, as the authors mention that this is an "open research-oriented" dataset.

**Reproducibility:**

4: Could mostly reproduce the results, but there may be some variation because of sample variance or minor variations in their interpretation of the protocol or method.

**Reviewer Confidence:**

4: Quite sure. I tried to check the important points carefully. It's unlikely, though conceivable, that I missed something that should affect my ratings.

---

> ### Author Rebuttal · Authors · 2023-08-29
>
> We would like to express our gratitude to Reviewer q3Sr for recognizing the potential of our work as a valuable resource in advancing research in the scientific domain. We also appreciate the identification of the typos, which will be addressed in the final manuscript.
>
> > Reject Reason A. *My only concern is that this corpus will be a source of unlabeled data. While the authors do provide topic classification labels for a small subset, it is not extendable without a significant amount of human effort. As a result, it is unclear how easy/feasible/suitable it will be for downstream applications.*
>
> Thanks for raising this pertinent concern. We would like to discuss certain promising methodologies for deriving labeled data beneficial for downstream applications, highlighting the corpus's potential versatility. In our topic classification demonstration, we propose an approach that leverages a model, fine-tuned on a limited amount of 'golden data', to predict 'silver topic labels' for the rest of unlabeled data in the OCL corpus. This 'golden data' can be curated through human annotations, ensuring precision. Given the demonstrated capabilities of large language models (LLM), we believe that LLMs can play an instrumental role in generating high-quality, large-scale silver labels [1]. Moreover, human-AI collaborative annotations [2] provide an effective strategy for complex tasks like natural language inference. We'll incorporate this discussion into the finalized version of the paper.
>
>
>
> > QA. *Did you try making use of Large Language Models to generate the keywords instead of using TF-IDF and Yake?*
>
> No, we did not. We initially focused on TF-IDF and Yake! due to their unsupervised approach, ease of use, and efficiency. Nevertheless, your suggestion holds merit. For example, employing prompt engineering to tailor Large Language Models (LLM), such as GPT-4 [3] or Llama2 [4], might offer more nuanced and high-quality keywords compared to the currently adopted methods. While we won't transit to LLM-based strategies in this version of the paper, we acknowledge the potential and will incorporate LLM into our topic classification system in subsequent iterations.
>
>
> References:
>
> [1] Yu, Danni, Luyang Li, Hang Su, and Matteo Fuoli. 2023. “Using LLM-Assisted Annotation for Corpus Linguistics: A Case Study of Local Grammar Analysis.” arXiv. http://arxiv.org/abs/2305.08339.
>
> [2] Liu, Alisa, Swabha Swayamdipta, Noah A. Smith, and Yejin Choi. 2022. “WANLI: Worker and AI Collaboration for Natural Language Inference Dataset Creation.” In Findings of the Association for Computational Linguistics: EMNLP 2022, 6826–47. Abu Dhabi, United Arab Emirates: Association for Computational Linguistics. https://doi.org/10.18653/v1/2022.findings-emnlp.508.
>
> [3] OpenAI. 2023. “GPT-4 Technical Report.” arXiv. http://arxiv.org/abs/2303.08774.
>
> [4] Touvron, Hugo, Louis Martin, Kevin Stone, Peter Albert, Amjad Almahairi, Yasmine Babaei, Nikolay Bashlykov, et al. 2023. “Llama 2: Open Foundation and Fine-Tuned Chat Models.” arXiv. http://arxiv.org/abs/2307.09288.

---

### Meta-Review · Area_Chair_SdZm · 2023-09-09

**Recommendation:** 5

**Metareview:**

This paper presents a scholarly corpus containing 73K peer reviewed publications from various computational linguistics conferences and workshops from the past seven decades. It massively extends upon existing corpora, whose contents may now be considered obsolete and includes topic labels for a subset of publications that may benefit downstream research on topic classification. Reviewers were pleased with the design of the corpus, the data collection and curation methodology (two 3s, two 4s for Soundness), and the inclusion of case studies demonstrating its utility and impact (e.g. the corpus may be used to observe waning and resurgence trends among particular topics). There were some concerns over the format used (Parquet) and the lack of labels for every item in the corpus, but these were addressed to reviewer satisfaction during the author rebuttal period. Lastly, though corpus creation is not particularly exciting research, the Excitement scores were still relatively high (two 3s, two 4s). One reviewer also suggested this submission would be better as a short paper, but I believe the authors’ defense to be sufficient. I would, however,  advise authors to consider extending their bullet point list of contributions to also include those mentioned in the rebuttal.

In summary, only minor revisions, addressing reviewers’ comments and questions, need to be made to ensure this paper is camera ready.

---

### Decision · Program_Chairs · 2023-10-07

**Decision:**

Accept-Main

**Comment:**

This paper presents a scholarly corpus containing 73K peer reviewed publications from various computational linguistics conferences and workshops from the past seven decades. It massively extends upon existing corpora, whose contents may now be considered obsolete and includes topic labels for a subset of publications that may benefit downstream research on topic classification. Reviewers were pleased with the design of the corpus, the data collection and curation methodology (two 3s, two 4s for Soundness), and the inclusion of case studies demonstrating its utility and impact (e.g. the corpus may be used to observe waning and resurgence trends among particular topics). There were some concerns over the format used (Parquet) and the lack of labels for every item in the corpus, but these were addressed to reviewer satisfaction during the author rebuttal period. Lastly, though corpus creation is not particularly exciting research, the Excitement scores were still relatively high (two 3s, two 4s). One reviewer also suggested this submission would be better as a short paper, but I believe the authors’ defense to be sufficient. I would, however,  advise authors to consider extending their bullet point list of contributions to also include those mentioned in the rebuttal.

In summary, only minor revisions, addressing reviewers’ comments and questions, need to be made to ensure this paper is camera ready.